# Intratumoral Genomic Heterogeneity May Hinder Precision Medicine Strategies in Patients with Serous Ovarian Carcinoma

**DOI:** 10.3390/diagnostics10040200

**Published:** 2020-04-03

**Authors:** Kohei Nakamura, Eriko Aimono, Shigeki Tanishima, Mitsuho Imai, Akiko Kawano Nagatsuma, Hideyuki Hayashi, Yuki Yoshimura, Kentaro Nakayama, Satoru Kyo, Hiroshi Nishihara

**Affiliations:** 1Genomics Unit, Keio Cancer Center, Keio University School of Medicine, 35 Shinanomachi, Shinjukuku, Tokyo 160-8582, Japan; eriko0123@keio.jp (E.A.); mitsuho.imai@keio.jp (M.I.); ak-nagatsuma@keio.jp (A.K.N.); rock-hayashi-pop@keio.jp (H.H.); hnishihara1971@keio.jp (H.N.); 2Department of Obstetrics and Gynecology, Kumagaya General Hospital, Saitama 360-8657, Japan; 3Department of Biomedical Informatics, Kansai Division, Mitsubishi Space Software Co., Ltd, Tokyo 661-0001, Japan; Tanishima.Shigeki@mss.co.jp; 4Department of Obstetrics and Gynecology, Shimane University School of Medicine, Enyacho 89-1, Izumo 693-8501, Japan; kiyu1225@yahoo.co.jp (Y.Y.); kn88@med.shimane-u.ac.jp (K.N.); satoruky@med.shimane-u.ac.jp (S.K.)

**Keywords:** ovarian carcinoma, genomic heterogeneity, precision medicine, DNA sequencing

## Abstract

Precision medicine, which includes comprehensive genome sequencing, is a potential therapeutic option for treating high-grade serous carcinoma (HGSC). However, HGSC is a heterogeneous tumor at the architectural, cellular, and molecular levels. Intratumoral molecular heterogeneity currently limits the precision of medical strategies based on the gene mutation status. This study was carried out to analyze the presence of 160 cancer-related genetic alterations in three tissue regions with different pathological features in a patient with HGSC. The patient exhibited histological heterogeneous features with different degrees of large atypical cells and desmoplastic reactions. TP53 mutation, ERBB2 and KRAS amplification, and WT1, CDH1, and KDM6A loss were detected as actionable gene alterations. Interestingly, the ERBB2 and KRAS amplification status gradually changed according to the region examined. The difference was consistent with the differences in pathological features. Our results demonstrate the need for sampling of the appropriate tissue region showing progression of pathological features for molecular analysis to solve issues related to tumor heterogeneity prior to developing precision oncology strategies.

## 1. Introduction

Epithelial ovarian cancer (EOC), a heterogeneous disease, comprises five main histologic types: high-grade serous (70%), clear cell (10%), endometrioid (10%), mucinous (3%), and low-grade serous carcinomas (<5%) [1]. Given that they display distinct histopathology, morphology, and genomic changes, the five EOC subtypes are regarded as unique diseases requiring careful diagnosis and well-defined treatments [2]. EOC subtypes can be categorized as type I and type II according to their molecular profile, disease progression, and prognosis [3]. Type I carcinomas develop slowly and comprise low-grade serous, low-grade endometrioid, mucinous, and clear cell carcinomas with mutations in *KRAS*, *BRAF*, and *PIK3CA* and a lack of *TP53* mutations [4]. In contrast, type II tumors are aggressive and characterized by high-grade serous carcinoma (HGSC), high-grade endometrioid, and undifferentiated carcinomas with mutations in *TP53* [5]. Type I and II tumors exhibit distinct expression clusters, with the latter characterized by genomic instability, severely aggressive clinical progression, and poor prognosis [6]. Although clear cell carcinoma is listed as a type I tumor, it may belong to an intermediate category because of its unique mutations and behavior.

Recent studies have revealed the germline alterations linked to cancer risk and somatic changes in HGSC [6,7]. Among analyzed HGSC tumor samples, 96% showed somatic TP53 mutations. Moreover, low-prevalence mutations were detected in nine other genes, including *NF1*, *BRCA1*, *BRCA2*, *RB1*, and *CDK12*. Additionally, the tumors exhibit large repeated copy number alterations (8 gains and 22 losses) involving genes such as *MYC* and *KRAS* (gains) and *PTEN*, *RB1*, and *NF1* (losses). Most tumors exhibit large gains and losses, highlighting the genomic instability of HGSC. 

Precision medicine, such as approaches involving comprehensive genome sequencing, is a potential treatment option for different types of cancer [8]. Pathologists have long acknowledged the existence of tumor morphological heterogeneities, which are used as a foundation for various prognostic classification systems for grading tumors. For example, in breast cancer, the Scarff–Bloom–Richardson grading system includes evaluation of nuclear pleomorphism, a feature linked to tumor aneuploidy. The word “anaplasia,” which was first used in 1890 by Von Hansemann, alludes to nuclear and mitotic atypia and indicates tumor morphological heterogeneity. Tumor morphological heterogeneity is typically unique to a region but varies in tumor cell proliferation, immune infiltration, differentiation status, and necrosis. Therefore, variations in the histological characteristics of cancers may be linked to known molecular intratumoral heterogeneity and different responses to treatment [9,10]. As a consequence, because of restrictions on the number of biopsy cores typically extracted in carcinoma mass biopsies, intratumoral molecular heterogeneity may limit the effectiveness of precision cancer therapies [11]. In the present study, we analyzed 160 cancer-related genes in three tissue regions from a patient with HGSC to identify the possible limitations of molecular research based on insufficient sampling of intratumoral heterogeneity. This information is useful for defining tumor sampling strategies to improve precision medicine methods.

## 2. Materials and Methods

### 2.1. Molecular Analysis

We investigated the presence of mutations in ~160 cancer-related genes in three tumor regions (Figure 1) of a right ovarian tumor. Sections (10 µm) were dissected to provide >50% tumor cells in the specimens and minimize the presence of necrosis. The genomic testing system used was an internal clinical sequencing apparatus named as “PleSSision”, which is used for all genome sequencing-related analyses in our hospital and from our collaborators, including Keio University Hospital. This apparatus was used to extract genomic DNA from tumor samples and peripheral blood mononuclear cells extracted from patients with cancer after their consent was obtained to undergo comprehensive genomic testing. All subjects gave their informed consent for inclusion before they participated in the study. The study was conducted in accordance with the Declaration of Helsinki, and the protocol was approved by the Ethics Committee of Shimane University Faculty of 46 Medicine, Izumo, Japan (no.: 960).

DNA quality was checked by calculating the DNA integrity number using the Agilent 2000 TapeStation (Agilent Technologies, Santa Clara, CA, USA) prior to targeted amplicon exome sequencing of 160 genes implicated in cancer using the Illumina MiSeq sequencing platform (Illumina, San Diego, CA, USA). The 160 genes examined are listed in Appendix A. All DNA samples analyzed had a DNA integrity number of at least 3.1. The sequencing data were entered into the GenomeJack bioinformatics pipeline (Mitsubishi Space Software, Tokyo, Japan) and analyzed within 3 days. We found cancer-specific changes in somatic genes, including single-nucleotide variations, insertions/deletions, and copy number variations. These findings were used to determine the tumor mutational burden. The annotated analysis findings were reviewed at a genome conference attended by medical professionals including medical oncologists, molecular oncologists, pathologists, medical geneticists, clinical laboratory technicians, bioinformaticians, genetic counselors, pharmacists, and nurses. The final report, which contained the suggested treatment according to the genomic profiling results, was finalized following approval by experts at this conference and was provided to physicians and patients. This process had an anticipated turnaround of approximately 2 weeks.

### 2.2. Immunohistochemical HER2 Assessment

Immunohistochemical *HER2* assessment was performed by medical technologists at our hospital according to ASCO/CAP guidelines.

### 2.3. Reporting of Secondary Germline Findings

In our system, the genomic profiles of tumor tissue and peripheral blood mononuclear cells were compared to identify secondary germline findings. Therefore, in the initial visit, we asked all patients to disclose the secondary germline findings from their genomic test. For patients who agreed, we only disclosed this information once we confirmed that the detected germline variants were pathogenic using a global cancer genome database such as ClinVar. Additionally, we abided by the ACMG’s recommendations for the reporting of secondary findings in clinical exome and genome sequencing studies.

## 3. Results

### 3.1. Patient and Pathological Findings

The patient analyzed in the present study was a 67-year-old woman undergoing a total abdominal hysterectomy, bilateral salpingo-oophorectomy, and omentectomy. An ovarian mass (right ovary) measuring 13 cm in length had metastasized to the uterus, sigmoid colon, rectum, and omentum. One of the metastatic tumors occupied approximately 20 cm of the omental cake’s width. In the omental cake, very few parts showed tumor cells with glandular or papillary formation. Many tumor cells were proliferative, with small to large solid nests. Although the polymorphisms of the tumor cells and desmoplastic reaction were mild (Figure 1a), the tumor cells had a high frequency of abnormal nuclei and we observed increased severity of the tumor’s fibrosis lesion progression (Figure 1b,c). The degrees of the desmoplastic reactions gradually increased from region a to c. For neoplastic cells, overexpression of p53 protein in all regions was observed by immunostaining. There was no evidence of serous intraepithelial carcinoma in the oviduct or fimbriae tube. A pathologist evaluated the resected samples and diagnosed the mass as HGSC (pT3NxM1).

### 3.2. Sequencing Results

Genomic DNA sequences were obtained from three regions (named as a, b, and c) classified as high-grade serous carcinomas (average sequencing depth of 538× (a) 567× (b) 592× (c)). The average tumor cellularity was 50% for the three samples, as determined by pathological review of each sample. Moreover, tumor cellularity was approximated based on variant allele frequencies of 80% (a), 60% (b), and 70% (c). The gene alteration profiles detected in each region are shown in Table 1. Functional gene alterations were found in each tumor sample. *TP53* mutation and *CDH1*/*KDM6A*/*WT1* loss were detected in all regions. Interestingly, *ERBB2* amplification was detected in regions b and c (copy number (CN) = 32 (b), and 26 (c)), whereas *ERBB2* was only slightly amplified in region a (CN = 6). Furthermore, *KRAS* amplification was detected in region c (CN = 4) but not in regions a and b. Among these gene alterations, *ERBB2* amplification is a potential druggable gene alteration. The phylogenetic tree summary is shown in Figure 2. In all, seven genes were affected; the remainder of the 160 genes did not contain any mutations or copy number alterations.

Next, we performed an immunohistochemical assessment for human epidermal growth factor receptor (*HER2*) in each region. The results were 1+ (Figure 3a) and 3+ (Figure 3b,c), and they were consistent with the results of the *ERBB2* amplification discussed in the previous section.

### 3.3. Secondary Germline Findings

Secondary germline findings were identified for the heterozygous *MUTYH* pathogenic variant G283E. *MUTYH* is listed in the ACMG recommendations, and the patient provided consent for disclosure of her germline information; therefore, the patient was informed of our findings. A heterozygous *MUTYH* variant may increase the risk of colorectal cancer but it is not associated with the risk of ovarian cancer.

## 4. Discussion

The genetic heterogeneity of HGSC has been described previously [12,13,14]. In 66% of cases, HGSC occurs bilaterally and often synchronously, affecting both ovaries; whether they are independent primary tumors from multifocal oncogenesis, arising spontaneously from a similar genetic background, clonally related due to tumorigenesis initiating from one ovary and then metastasizing to the contralateral ovary, or two metastases, has also been evaluated [15]. Although these studies revealed clonal relationships between the primary and metastatic tumors, limited information is available on the genetic heterogeneity in the primary tumor region and the influence of histological differences.

We evaluated the presence of mutations in 160 cancer-related genes in three tissue regions of a patient diagnosed with HGSC. A *TP53* mutation and *CDH1*/*KDM6A*/*WT1* loss were detected in all regions. Interestingly, the *ERBB2* amplification status gradually changed from region a to region b. Furthermore, *KRAS* amplification was newly detected only in region c. The results emphasize the presence of intratumoral molecular heterogeneity, which may hinder the application of personalized-medicine approaches given that molecular analysis for diagnosis is typically conducted on one tumor sample extracted from patients with HGSC. Our findings emphasize the need to improve sampling procedures for molecular analysis of HGSCs. Therapeutic decision-making in oncology is frequently based on a single tumor lesion. This method is expected to be therapeutically tractable if tumor somatic events arise in the tumor trunk and occur universally in all tumor subclones, and continually sustain tumor growth and survival at all locations. However, temporal and spatial alterations to tumor subclonal architecture dynamics may cause sub-dominant clones, such as those that were absent or barely detectable at the primary location, to develop pre-eminence. Therefore, variations in tumor environmental selection pressure, even at primary tumor locations, can cause regional differences in tumor subclone evolution, where each environment selects for one subclone over another, leading to additional intratumoral genetic heterogeneity. As a consequence, changes to a tumor’s subclonal architecture may lead to differences in tumor molecular profiles within a single primary tumor location. As evidenced in our study, sampling of an appropriate tumor region, which appears to be progressing with respect to pathological features such as high nuclear grade or high degree of desmoplastic reactions, may enable identification of foci containing mutations, particularly when some histological features are observed in one sample. Broad sampling may be useful but requires increased cost and effort. These results agree with those of Ruiz-Cerda [16] who highlighted the importance of the sample size when analyzing this type of data, and those of Jiang, [17] who observed that intratumoral heterogeneity is prevalent in renal cell carcinoma.

We also identified an intriguing relationship between the tumors’ histological characteristics and presence/absence of the mutation or amplification of interest. In this case, *KRAS* or *ERBB2* was gradually amplified as the cancer progressed because the area in which the desmoplastic reaction was low did not show *KRAS* or *ERBB2* amplification, and that in which the desmoplastic reaction was high exhibited amplification of these genes. In some cases, we observed different histological features such as nuclear grade 1, 2, and 3 or the desmoplastic reaction. In these cases, the cancer initially developed as nuclear grade 1 before progressing to grade 2 or 3 carcinoma. Our results suggest that genetic alterations are added as cancer progresses. A careful review of a tumor’s morphological features is useful for predicting the existence of mutations or amplification, indicating that this strategy is effective for determining the most suitable tissue areas for molecular analysis. Using only sampled one region (e.g., region a), we would not have observed the amplification of *KRAS* or *ERBB2*, and therefore would not have been able to provide a precise HER2-targeted therapy. Therefore, we recommend sampling each region with a different histological status when performing comprehensive genome sequencing. 

The genetic pathways involved in neoplasia have gained increased attention because of the promising developments in targeted therapies, thereby improving the potential of personalized medicine strategies. Previous studies revealed several genetic abnormalities in oncogenes and tumor suppressor genes implicated in HGSC tumorigenesis. Among these, the most common were TP53 mutations, occurring in approximately 96% of cases and activating mutations of the retinoblastoma pathway [6]. RAS/PI3K signaling is the main pathway implicated in high-grade serous ovarian cancer and is disrupted in 45% of tumors [6]. The RAS/PI3K pathway is comprised of the RAS/ERK and PI3K/AKT pathways, each of which are affected by distinct genetic components, including KRAS and BRAF lesions or PTEN and PIK3CA mutations, respectively [6]. 

A study by Gerlinger et al. [18] examined intratumoral heterogeneity by exome sequencing of diverse tissue regions and revealed variation in driver mutations and driver copy numbers among the different regions. The same study identified a driver mutation of PI3K in 4% of cases using a single-biopsy method compared to 20% of cases using multiregional sequencing. This demonstrates that single-biopsy methods underestimate the prevalence of driver mutations in renal cell carcinoma. Moreover, a study by Martinez et al. [19] indicated that copy number aberrations contribute to both intertumoral and intratumoral heterogeneity in renal cell carcinoma and may cluster more with other renal cell carcinomas than with subclones within the same lesion. Therefore, efficient and extensive tumor sampling along with broad molecular analyses appears to be useful for not only determining the diagnosis and stage of disease but also recognizing relevant genetic abnormalities. Sampling biases may, therefore, be largely responsible for the absence of clinically qualified biomarkers for HGSC.

There were two main limitations to this study. First, as only one case study was examined, we could not provide substantial evidence to indicate that the differences in histological features were associated with differences in genetic alterations. Analyzing further case studies is necessary to fully understand this association. Second, our study investigated only 160 cancer-related genes. If the whole genome had been sequenced and all cancer-related genes analyzed, further information would have been revealed.

In conclusion, the three tissue regions examined showed different microscopic features, including the desmoplastic reaction, and correlations of the mutational status with histological characteristics. Our findings emphasize the need for sampling of appropriate tissue regions showing progression of pathological features for molecular analysis to solve data analysis problems and understand intratumoral heterogeneity in HGSC for developing precision medicine strategies.

## Figures and Tables

**Figure 1 diagnostics-10-00200-f001:**

High-grade serous carcinoma with cells with glandular, papillary, and solid arrangement. Desmoplastic reaction gradually increasing from region (**a**) (left panel) → (**b**) (middle panel) → (**c**) (right panel) (×200).

**Figure 2 diagnostics-10-00200-f002:**
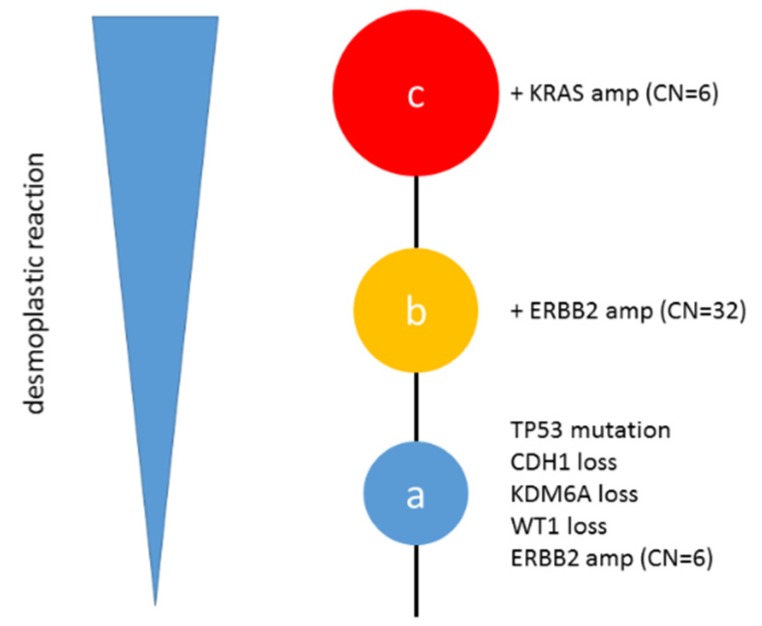
Phylogenetic tree summary. Phylogenetic tree for this tumor from region a to c.

**Figure 3 diagnostics-10-00200-f003:**
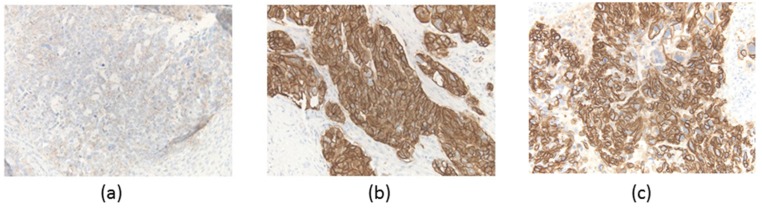
Immunohistochemical *HER2* assessment. Region (**a**) (left panel): 1+, region (**b**) (middle panel): 3+, region (**c**) (right panel) (×200).

**Table 1 diagnostics-10-00200-t001:** Detailed information regarding actionable gene alterations in three tumor regions.

Area	Actionable Gene Alterations	Tumor Mutation Burden (SNVs/Mbp)
a	*TP53* V272M, *ERBB2* amp (CN = 6), *CDH1* loss, *KDM6A* loss, *WT1* loss	2.7
b	*TP53* V272M, *ERBB2* amp (CN = 32), *CDH1* loss, *KDM6A* loss, *WT1* loss	2.7
c	*TP53* V272M, *ERBB2* amp (CN = 26), *CDH1* loss, *KDM6A* loss, *WT1* loss, *KRAS* amp (CN = 4)	2.7

CN: copy number; SNVs: single-nucleotide variants.

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
