# Peer review of "Intratumoral Genomic Heterogeneity May Hinder Precision Medicine Strategies in Patients with Serous Ovarian Carcinoma"

_diagnostics, 2020, doi:10.3390/diagnostics10040200_

Round 1
Reviewer 1 Report
1) Table 1: Please consider including the full panel results and describing and analyzing differences between regions more extensively. See Masoodi etal Br J Cancer 2020; Feb 26 for an example of how the data could be presented in a more sophisticated manner.
2) Overall: I think the conclusion and discussion regarding a need for sampling of broader regions are overstated for based on data presented in this paper. There is only one patient, only 160 genes are evaluated and they are not all that different. Please consider whether one case really supports the extra cost and effort required for multiple region sampling and revise abstract, discussion and conclusions accordingly.
3) Discussion: A better discussion about what is known about heterogeneity in ovarian cancer is needed, and how this paper fits into the field.
See Masoodi, cited above
Takaya H et al, Gynecol Oncol 2020 156: 415-22
O'Donnell RL et al Int J Gynecol Cancer 2016;1004-11.
Author Response
Thank you for reviewing our manuscript and providing valuable comments. Our responses are shown below.
Q1) Table 1: Please consider including the full panel results and describing and analyzing differences between regions more extensively. See Masoodi et al Br J Cancer 2020; Feb 26 for an example of how the data could be presented in a more sophisticated manner.
Response A1) Thank you for your comment. According to your suggestions, we have provided additional information on the genetic test in the three regions and the figure and relevant sentence have been revised (page 4 , lines 138-43 and Table 1). Furthermore, we have added a phylogenetic tree summary of genetic alteration to the figure for easy visual inspection (page 5, Figure 3). We have also cited the suggested reference (page 5, lines 164).
Q2) Overall: I think the conclusion and discussion regarding a need for sampling of broader regions are overstated for based on data presented in this paper. There is only one patient, only 160 genes are evaluated and they are not all that different. Please consider whether one case really supports the extra cost and effort required for multiple region sampling and revise abstract, discussion and conclusions accordingly.
Response A2) Thank you for your suggestion. We agree with your comment. Broad sampling will require extra-cost and effort, and this may be an overstatement and unrealistic in the clinic. We provided one option for sampling of the appropriate tissue region appearing to show progression of pathological features such as a high nuclear grade or high degree of desmoplastic reaction. This may enable identification of foci containing mutations, particularly when some histological features are observed in one sample. Accordingly, we have modified the sentence in the Abstract, Discussion, and Conclusions (page 1, line 33-35, page 6, line 189-193, 208-209, 237-241).
Q3) Discussion: A better discussion about what is known about heterogeneity in ovarian cancer is needed, and how this paper fits into the field.
See Masoodi, cited above
Takaya H et al, Gynecol Oncol 2020 156: 415-22
O'Donnell RL et al Int J Gynecol Cancer 2016;1004-11.
Response A3) Thank you for your comment. According to your suggestion, we have added information on the genetic heterogeneity in HGSC to the Discussion section and cited the suggested reference (page 5, lines 164-171).
Reviewer 2 Report
Dear Authors,
thank you for the oportunity to review your work. Altought the issue of the study is important, im my oppinion, the importance of your study is poor. That is due to limited number of cases. You have evaluated TWO (or three? See below) samples of ONE tumor (one patient). In this situation, you paper should be regarded as a case-study. It is impossible to develope important conclusions on the basis on such limited data. In my oppinion, at least 30 cases is required.
Some minior comments are listed below
1) the discussion section is inadequate. The issue about genetic heterogenicity is broad, howerver, in my oppinion the discussion section report the issue insuficciently.
2) significant English edidting is required
3) You have written, that you performed the evaluation of 160 genes, however, you have mention only the few of them.
46 and clear cell
47 carcinomas with mutations in KRAS, BRAF, and PIK3CA and a lack of TP53 mutations
Nowel data sugest that true clear cell carcinomas cannot be classified as type I EOC. Please check and change it.
27 mutation status. Methods: This study was carried out to analyze the presence of 160 cancer-related
28 genetic alterations in two tissue regions with different pathological features from this case of HGSC.
78 We investigated the presence of mutations in ~160 cancer-related genes in three tumor regions
79 (Figure 1) of a right ovarian tumor
132 Genomic DNA sequences were obtained from two regions (called a, and b) classified as high133
grade serous carcinomas
So how many samples from one tumor did you evaluated? Two or three?
Author Response
Thank you for reviewing our manuscript and providing valuable comments. Our responses are shown below.
(comment)
Thank you for the opportunity to review your work. Although the issue of the study is important, in my opinion, the importance of your study is poor. That is due to limited number of cases. You have evaluated TWO (or three? See below) samples of ONE tumor (one patient). In this situation, your paper should be regarded as a case-study. It is impossible to develop important conclusions on the basis on such limited data. In my opinion, at least 30 cases is required.
Response A1) Thank you for your comment. We agree that the number of cases is small. We evaluated three samples from one patient. According to your suggestion, we have changed the category of the manuscript from ‘original article’ to ‘case report’.
(Minor comment)
Q1) The discussion section is inadequate. The issue about genetic heterogeneity is broad, however, in my opinion the discussion section report the issue insufficiently.
Response A1) Thank you for your comment. Based on your suggestion, we have added information on the genetic heterogeneity in HGSC, which was determined in previous studies, to the Discussion (page 5, lines 164-171).
Q2) Significant English edidting is required
Response A2) The manuscript has been revised by a native English speaker working for ‘Editage’. We have attached the certification of English editing with the revised manuscript.
Q3) You have written, that you performed the evaluation of 160 genes, however, you have mention only the few of them.
Response A3) We have included only information on genes with mutation or copy number alterations. We modified the sentence to clarify this point (page 4, lines 133-143). Furthermore, we have added information on the 160 genes examined in this genomic test to Supplemental Table S1.
Q4) and clear cell carcinomas with mutations in KRAS, BRAF, and PIK3CA and a lack of TP53 mutations (Line 46-47)
Novel data suggest that true clear cell carcinomas cannot be classified as type I EOC. Please check and change it.
Response A4) As you pointed out, clear cell carcinoma is listed as a type I tumor, however, it may actually belong to an intermediate category because of its mutations and behavior. We have added this information to the revised manuscript (page 2, lines 50-51).
Q5) mutation status. Methods: This study was carried out to analyze the presence of 160 cancer-related genetic alterations in two tissue regions with different pathological features from this case of HGSC. (Line 27-28) We investigated the presence of mutations in ~160 cancer-related genes in three tumor regions (Figure 1) of a right ovarian tumor (Line 78-79) Genomic DNA sequences were obtained from two regions (called a, and b) classified as high grade serous carcinomas (Line 132-133)
So how many samples from one tumor did you evaluated? Two or three?
Response A5) Thank you for your comment. We apologize for the confusion. We have analyzed the genetic testing in three regions of one patient but provided only two figures to simplify the results. Based on your comments, we have included all information from the three regions in the revised manuscript and modified the sentence and figures (page 4, lines 133-146).
Round 2
Reviewer 1 Report
Revised approrpiately
Author Response
Thank you for reviewing our revised manuscript.
Reviewer 2 Report
The paper improved significantly, however, due to limited number of cases, I consider its priority as low.
Few issues needs clarification.
1) you have examined 160 gens, however, you have present the results of only few of them. What about the rest? If there were no mutation or there were no differences between the regions, please write it clearly.
2) in my opinion, the association between the ERBB2 and KRAS mutations with desmoplastic reaction is very speculative. In my oppinion, it is very dangerous to suggest the association that is based on the examination of only one tumor. I would suggest to abandon this issue.
Author Response
Thank you for reviewing our revised manuscript and providing valuable comments. Our responses are shown below.
Q1) you have examined 160 gens, however, you have present the results of only few of them. What about the rest? If there were no mutation or there were no differences between the regions, please write it clearly.
Response 1) Thank you for your comment. We included only information on the few genes with mutations or copy number alterations; the remaining genes did not contain any mutations. Per your suggestion, we have modified a sentence to clarify this point (page 4, lines 143–144).
Q2) in my opinion, the association between the ERBB2 and KRAS mutations with desmoplastic reaction is very speculative. In my opinion, it is very dangerous to suggest the association that is based on the examination of only one tumor. I would suggest to abandon this issue.
Response 2) Thank you for your valid concern. Because only one case study was examined, we could not provide substantial evidence to indicate that the differences in histological features were associated with differences in genetic alterations. We had stated this limitation in the previously submitted revised version of the manuscript (page 6, lines 233–235 of the current version).